# Cross-Domain Validation of a Resection-Trained Self-Supervised Model on Multicentre Mesothelioma Biopsies

**Farzaneh Seyedshahi**[1,2] **Francesca Damiola**[3] **Sylvie Lantuejoul**[3,4] **Ke Yuan**[1,2] **John LeQuesne**[1,2,5]

F.SEYEDSHAHI.1@RESEARCH.GLA.AC.UK

[1] *School of Cancer Sciences, University of Glasgow, Glasgow, Scotland, UK*

[2] *Cancer Research UK Scotland Institute, Glasgow, Scotland, UK*

[3] *Department of Biopathology, Léon Bérard Cancer Center, Lyon, France*

[4] *Grenoble Alpes University, Grenoble, France*

[5] *Queen Elizabeth University Hospital, NHS Greater Glasgow and Clyde, Glasgow, Scotland, UK*

## Abstract

Accurate subtype classification and outcome prediction in mesothelioma are essential for guiding therapy and predicting patient outcomes. However, most computational pathology models are trained exclusively on large tissue images from resection specimens, which limits their relevance in real-world diagnostic settings where small biopsies are the primary tissue source. Here, we assess the biopsy-level generalisability of a self-supervised encoder using a large, multicentre French cohort. We identify 53 biopsy-specific histomorphological clusters, quantify each patient's proportional representation across these clusters, and use these profiles as inputs for two downstream tasks. The results show that a self-supervised encoder trained on resection tissue can be reliably transferred to biopsy material despite significant domain shifts.

**Keywords:** Mesothelioma, Biopsy, Self-supervised Learning, Histopathology.

## 1. Introduction

Mesothelioma is a rare and aggressive cancer characterised by substantial histological heterogeneity and poor prognosis, where accurate subtype classification and survival risk stratification are essential for clinical decision-making (Wagner et al., 1960; Molinari). However, conventional histopathological diagnosis remains subjective and often exhibits high inter-observer variability, particularly in cases with transitional morphological patterns(Travis et al., 2015; Salle et al., 2018; Scherpereel et al., 2020). While recent computational pathology methods, including deep learning and graph-based models (Courtiol et al., 2019; Eastwood et al., 2023), have shown promise in improving prognostic prediction and subtype characterisation, many approaches rely on weak supervision using human-provided labels. In mesothelioma, where morphological patterns are often ambiguous and pathologist disagreement is common, such dependence risks propagating label noise and limiting the robustness of learned representations. Although self-supervised learning has emerged as a promising alternative by learning representations directly from unlabeled data, most existing models are trained on large resections rather than the small biopsy samples used in routine diagnosis (Chen et al., 2024; Vorontsov et al., 2024; Seyedshahi et al., 2025). This

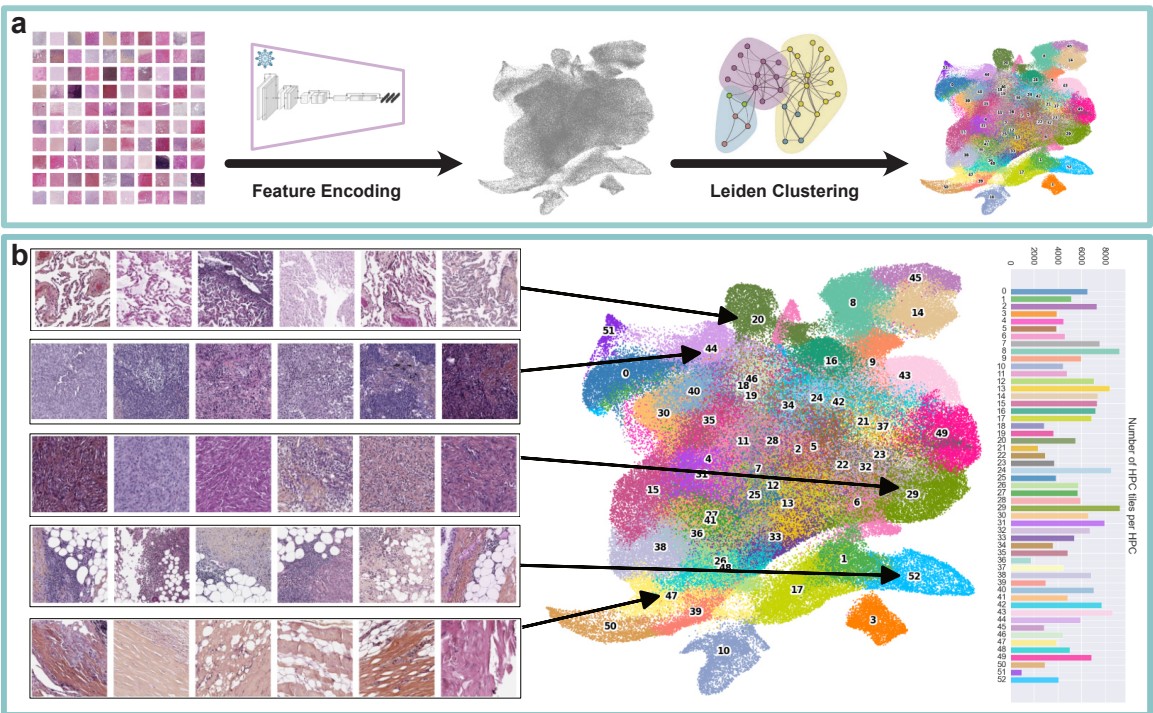

Figure 1: **(a)** Overview of the analysis pipeline **(b)** Each histomorphological phenotype cluster (HPC) shows a distinct and internally consistent morphological pattern.

mismatch between development data and clinical material introduces a significant gap in applicability. Motivated by this limitation, we evaluate whether a self-supervised histomorphological encoder trained on resection specimens can reliably transfer to real-world biopsy material despite substantial domain shifts in tissue type, staining protocols, and acquisition settings, and whether the resulting biopsy-derived representations retain diagnostically and prognostically meaningful information for mesothelioma analysis.

## 2. Methodology

To investigate whether representations learned from large surgical resections can transfer to real-world biopsy material, we design a controlled experimental framework focused on mesothelioma biopsies. This setting allows us to isolate the effects of domain shift between resection-trained encoders and biopsy slides, which differ substantially in tissue composition, staining protocols, and sampling depth. Using a large multicentre biopsy cohort, we apply a pretrained self-supervised encoder (Seyedshahi et al., 2025) without further fine-tuning to extract tile-level representations. By re-clustering these representations directly within the biopsy domain and evaluating their diagnostic and prognostic relevance, this framework provides a systematic assessment of whether self-supervised morphological features trained on resections can retain meaningful clinical signals. (Figure 1a)

## 3. Results

Applying the pretrained encoder to the biopsy cohort and clustering the resulting tile-level embeddings with the Leiden algorithm revealed 53 distinct histomorphological phenotype clusters (HPCs). Despite being trained only on H&E resection slides, the encoder produced stable and discriminative representations on HES/HPS-stained biopsies, indicating robustness to domain shift. The resulting HPCs correspond to meaningful morphological patterns, including lung-like architectures, dense tumour cellularity, adipose tissue, and collagen-rich stromal regions (Figure 1b).

To evaluate the clinical utility of these biopsy-derived HPCs, we predicted patient survival. Using the pathological grading as a baseline (Pathologists diagnosis of the cases) resulted in relatively low predictive performance, with C-index values of 0.51 for the training resection cohort and 0.60 for both biopsy train and test sets. In contrast, applying a Cox proportional hazards model to the proportional representation of HPCs improved performance, achieving a C-index of 0.67 on the training set and 0.65 on the biopsy test set.

We further evaluated the discrimination between epithelioid and non-epithelioid biopsies (Subtype task). Logistic regression applied to HPC proportion features achieved a 5-fold cross-validated AUC of $0.92 \pm 0.03$ on the biopsy test set, while a representative multiple instance learning (MIL) baseline, CLAM (Lu et al., 2021), using the same encoder achieved $0.90 \pm 0.05$.

Table 1: Benchmarking the model's performance across the datasets.

| Methods | Task | Resection | Biopsy-Train | Biopsy-Test |
|---------|------|-----------|--------------|-------------|
| Pathological Diagnosis | Survival | $0.51 \pm 0.03$ | $0.60 \pm 0.01$ | $0.60 \pm 0.03$ |
| HPCs + Cox model | Survival | $0.65 \pm 0.03$ | $0.67 \pm 0.00$ | $0.65 \pm 0.02$ |
| CLAM (Lu et al., 2021) | Subtype | $0.87 \pm 0.04$ | $0.96 \pm 0.01$ | $0.9 \pm 0.05$ |
| HPCs + Logistic regression | Subtype | $0.88 \pm 0.04$ | $0.96 \pm 0.01$ | $0.92 \pm 0.03$ |

Overall, these results suggest that the unsupervised HPCs capture prognostically relevant information beyond traditional grading criteria and a classic MIL method. Future work will focus on biological interpretation of the HPCs and comparing with more baselines methods.

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
