# OpenReview forum: "Cross-Domain Validation of a Resection-Trained Self-Supervised Model on Multicentre Mesothelioma Biopsies"
_MIDL.io/2026/Short_Papers — MIDL 2026 - Short Papers Poster_

### Official Review · Reviewer_EtDp · 2026-05-04
**Interesting Domain Shift Study With Insufficient Validation Evidence**

**Rating:** 2
**Confidence:** 3

**Review:**

The clinical motivation is relevant, as biopsy material is closer to routine diagnostic practice than resection tissue. However, the presented results remain limited. For survival prediction, the biopsy-test C-index is 0.62 for the proposed cluster-based Cox model compared with 0.61 for pathological grade, which does not convincingly demonstrate added value over manual pathological assessment. For subtype classification, performance is high but only marginally better than the CLAM baseline. Overall, the study is best interpreted as a small preliminary validation experiment, not as evidence that the method reliably transfers across domains or improves on pathologist-derived information.

**Summary:**

This short paper investigates whether a self-supervised mesothelioma encoder trained on resection specimens can be transferred to biopsy material. The authors apply the frozen encoder to multicentre biopsy slides, recluster tile-level features into 53 histomorphological phenotype clusters, aggregate these clusters into patient-level proportions, and use these features for survival prediction and subtype classification.

**Strengths:**

- The paper addresses a clinically relevant domain-shift problem: transferring models trained on resection specimens to biopsy material.
- The validation setting is relevant because biopsy tissue is closer to routine diagnostic practice than resection material.
- The use of cluster proportions as patient-level features is potentially interesting and may offer a more interpretable alternative to purely black-box slide-level prediction.
- The study has a clear and compact experimental setup that is suitable for a short paper: frozen encoder, biopsy-domain clustering, patient-level aggregation, and evaluation on survival and subtype classification.

**Weaknesses:**

-	The main claims are too strong for the evidence shown. The results support, at most, a preliminary feasibility or validation study, not reliable cross-domain transfer from resection tissue to biopsy material.
-	The method does not convincingly improve over pathologist-derived assessment. For survival prediction, the authors state that the pathological grade baseline “resulted in relatively low predictive performance,” but the proposed method is not much higher on the biopsy test set. The C-index is 0.62 for the cluster-based Cox model compared with 0.61 for pathological grade, so the paper does not show clear added value over manual pathological information.
-	The baseline using pathological grade is insufficiently described.
-	The morphological interpretation of the clusters is not sufficiently validated. The authors state that the clusters represent meaningful histomorphological patterns, but it is unclear what reference standard was used.
-	The dataset description is lacking. The paper mentions a large multicentre French biopsy cohort, but does not provide essential information such as number of patients, number of slides, number of centres, subtype distribution, etc.

**Justification Of Rating:**

I recommend a weak reject because the paper addresses a relevant cross-domain validation question, but the presented results do not convincingly support the main claims of reliable transfer or added clinical value over pathologist-derived information. In particular, the proposed survival model performs only marginally better than the pathological-grade baseline, while key details on the dataset, baseline construction, and validation of the morphological clusters are missing.

---

### Decision · Program_Chairs · 2026-05-08

Accept (Poster)